# Use of Venetoclax in Patients with Relapsed or Refractory Acute Myeloid Leukemia: The PETHEMA Registry Experience

**DOI:** 10.3390/cancers14071734

**Published:** 2022-03-29

**Authors:** Jorge Labrador, Miriam Saiz-Rodríguez, Dunia de Miguel, Almudena de Laiglesia, Carlos Rodríguez-Medina, María Belén Vidriales, Manuel Pérez-Encinas, María José Sánchez-Sánchez, Rebeca Cuello, Alicia Roldán-Pérez, Susana Vives, Gonzalo Benzo-Callejo, Mercedes Colorado, María García-Fortes, María José Sayas, Carmen Olivier, Isabel Recio, Diego Conde-Royo, Álvaro Bienert-García, María Vahi, Carmen Muñoz-García, Cristina Seri-Merino, Mar Tormo, Ferran Vall-llovera, María-Ángeles Foncillas, David Martínez-Cuadrón, Miguel Ángel Sanz, Pau Montesinos

**Affiliations:** 1Hematology Deparment, Hospital Universitario de Burgos, 09006 Burgos, Spain; 2Research Unit, Fundación Burgos por la Investigación de la Salud (FBIS), Hospital Universitario de Burgos, 09006 Burgos, Spain; msaiz@hubu.es; 3Hematology Deparment, Hospital Universitario de Guadalajara, 19002 Guadalajara, Spain; duniamll@hotmail.com; 4Hematology Deparment, Hospital Universitario Puerta de Hierro, 28222 Madrid, Spain; a.laiglesia@gmail.com; 5Hematology Deparment, Hospital de Gran Canaria Dr. Negrin, 35010 Las Palmas de Gran Canaria, Spain; hematocritico@yahoo.es; 6Hematology Deparment, Hospital Universitario de Salamanca, 37007 Salamanca, Spain; mbvidriales@saludcastillayleon.es; 7Hematology Deparment, Hospital Clínico Universitario de Santiago de Compostela, 15706 Santiago de Compostela, Spain; m.p.encimas@gmail.com; 8Hematology Deparment, Hospital Universitario Lucus Augusti, 27003 Lugo, Spain; maria.jose.sanchez.sanchez3@sergas.es; 9Hematology Deparment, Hospital Clínico Universitario de Valladolid, 47003 Valladolid, Spain; rcuellogarcia@gmail.com; 10Hematology Deparment, Hospital Universitario Infanta Sofía, 28703 Madrid, Spain; aroldanp@salud.madrid.org; 11Hematology Deparment, Hospital Germans Trias i Pujol-ICO, 08907 Badalona, Spain; svives@iconcologia.net; 12Hematology Deparment, Hospital Universitario de La Princesa, 28006 Madrid, Spain; gonzalo.benzo@salud.madrid.org; 13Hematology Deparment, Hospital Universitario Marqués de Valdecilla, 39008 Santander, Spain; mercedes.colorado@scsalud.es; 14Hematology Deparment, Hospital Universitario Virgen de la Victoria, 29010 Málaga, Spain; mgarciafortes@gmail.com; 15Hematology Deparment, Hospital Universitario Doctor Peset, 46017 Valencia, Spain; sayas_mjo@gva.es; 16Hematology Deparment, Hospital General de Segovia, 40002 Segovia, Spain; molivier@saludcastillayleon.es; 17Hematology Deparment, Complejo Asistencial de Ávila, 05071 Ávila, Spain; irecio@saludcastillayleon.es; 18Hematology Deparment, Hospital Universitario Príncipe de Asturias, Alcalá de Henares, 28805 Madrid, Spain; diegoconderoyo@gmail.com; 19Hematology Deparment, Hospital Universitario de Canarias, 38320 Santa Cruz de Tenerife, Spain; varo_bienert@hotmail.com; 20Hematology Deparment, Hospital Universitario Virgen de Valme, 41014 Sevilla, Spain; mariavahi@gmail.com; 21Hematology Deparment, Hospital Universitario Virgen Macarena, 41009 Sevilla, Spain; camugar@gmail.com; 22Hematology Deparment, Hospital Central de la Defensa Gómez Ulla, 28047 Madrid, Spain; csermer@mde.es; 23Hematology Deparment, Hospital Clínico Universitario de Valencia, 46010 Valencia, Spain; tormo_mar@gva.es; 24Hematology Deparment, Hospital Universitari Mutua Terrasa, 08221 Barcelona, Spain; fvall_llovera@mutuaterrassa.cat; 25Hematology Deparment, Hospital Universitario Infanta Leonor, 28031 Madrid, Spain; mariaangeles.foncillas@salud.madrid.org; 26Hematology Deparment; Hospital Universitari I Politécnic La Fe, 46026 Valencia, Spain; martinez_davcua@gva.es (D.M.-C.); miguel.sanz@uv.es (M.Á.S.)

**Keywords:** venetoclax, acute myeloid leukemia, relapsed, refractory

## Abstract

**Simple Summary:**

The use of venetoclax combined with hypomethylating agents or low-dose cytarabine in patients with newly diagnosed acute myeloid leukemia unfit for intensive chemotherapy was recently approved. However, the evidence in relapse or refractory patients is still scarce. The cohort of patients included in our study was heavily pretreated and had a poor performance status. It is still necessary to identify those patients at higher risk of early death who would not benefit from this type of treatment. For these ultra-high-risk patients, other treatment strategies should be followed.

**Abstract:**

The effectiveness of venetoclax (VEN) in relapsed or refractory acute myeloid leukemia (RR-AML) has not been well established. This retrospective, multicenter, observational database studied the effectiveness of VEN in a cohort of 51 RR-AML patients and evaluated for predictors of response and overall survival (OS). The median age was 68 years, most were at high risk, 61% received ≥2 therapies for AML, 49% had received hypomethylating agents, and ECOG was ≥2 in 52%. Complete remission (CR) rate, including CR with incomplete hematological recovery (CRi), was 12.4%. Additionally, 10.4% experienced partial response (PR). The CR/CRi was higher in combination with azacitidine (AZA; 17.9%) than with decitabine (DEC; 6.7%) and low-dose cytarabine (LDAC; 0%). Mutated NPM1 was associated with increased CR/CRi. Median OS was 104 days (95% CI: 56–151). For the combination with AZA, DEC, and LDAC, median OS was 120 days, 104 days, and 69 days, respectively; *p* = 0.875. Treatment response and ECOG 0 influenced OS in a multivariate model. A total of 28% of patients required interruption of VEN because of toxicity. Our real-life series describes a marginal probability of CR/CRi and poor OS after VEN-based salvage. Patients included had very poor-risk features and were heavily pretreated. The small percentage of responders did not reach the median OS.

## 1. Introduction

Treatment options for relapsed or refractory acute myeloid leukemia (R/R- AML) are limited, and the outcome remains very poor [1,2]. Apart from including these patients in a clinical trial whenever it is possible, there is no consensus regarding the optimal approach to manage R/R-AML.

Early results of phase 1b/2 trials with venetoclax-based combinations in treatment naïve older AML patients were impressive. They showed a higher response rate and longer response duration, resulting in higher OS in these patients, compared to the historical rates with low-dose cytarabine (LDAC) or hypomethylating agents (HMAs) in monotherapy [3,4,5,6,7]. The superiority of the venetoclax combination was confirmed in two phase 3 trials comparing 5-azacitidine (AZA) or LDAC plus either venetoclax or placebo in newly diagnosed AML ineligible for intensive chemotherapy [8,9]. The exciting results with venetoclax in untreated unfit AML led to their off-label use in the R/R-AML setting. However, the evidence in R/R-AML patients is still scarce, and available data are mostly from single-center studies [10,11,12,13].

The aim of our study was to retrospectively analyze the effectiveness of off-label use of venetoclax combined therapy in patients with R/R-AML reported to the Programa Español para el Tratamiento de las Hemopatías Malignas (PETHEMA) epidemiologic registry.

## 2. Materials and Methods

### 2.1. Study Design and Patients

The PETHEMA AML registry (NCT02607059) includes patients diagnosed with AML, regardless of previous treatments. We retrospectively collected the main patient characteristics, treatment approaches, and outcomes. Demographic (age, sex), cytomorphologic data confirming AML diagnosis (according to the local routine practice), cytogenetics, description of the front-line treatment approach, disease response, and follow-up (relapse o death) were registered. Other clinical information, including baseline physical examination and laboratory test results, prior neoplastic/hematologic diseases, molecular markers, treatment-related toxicities, consolidation and post-consolidation schedules (e.g., transplant, maintenance), and hospital stays were registered and analyzed. All patient’s clinical data and disease baseline characteristics were registered at the time of diagnosis (before starting treatment).

### 2.2. Eligibility

Patients aged ≥ 18 years old at the time of AML diagnosis (according to the WHO 2008 criteria) that received venetoclax alone or in combination with other drugs for R/R-AML were eligible for this retrospective analysis (PET-VEN-2020-01 study). Patients with newly diagnosed AML or with acute promyelocytic leukemia were excluded. Patients treated with up-front venetoclax-based strategies for AML were also excluded.

Informed consent was a requisite for patients alive. The protocol complied with the current Spanish legislation on clinical research in humans and was approved by the corresponding Research Ethics Board according to the Declaration of Helsinki.

### 2.3. Treatment Schedules

Venetoclax was administered for R/R-AML in combination with HMAs (azacitidine 75 mg/m^2^ subcutaneously daily for 7 days, or decitabine 20 mg/m^2^ intravenously for 5 days of a 28-day treatment cycle) or LDAC 20 mg/m^2^ subcutaneously once daily on days 1 to 10 every 28 days, according to investigator choice.

The dose of venetoclax was adjusted based on combination therapy (400 mg with HMA or 600 mg with LDAC) and concomitant medication with CYP3A4 inhibitors (such as voriconazole or posaconazole) [14]. Therefore, the concomitant medication data were also collected, especially strong and moderate CPY3A4 inhibitors, with the aim of assessing overexposure to venetoclax treatment and its possible relationship with adverse events.

### 2.4. Study Definitions and Variables

As this was a retrospective study, all data were collected from the patient’s recorded clinical history.

A patient was considered to be relapsed when, after reaching complete remission (CR) including CR with incomplete hematological recovery (CRi), presented at least one of the following: (i) reappearance of leukemic blasts in peripheral blood (PB), confirmed by a count of ≥5% blasts in bone marrow (BM), not attributable to any other cause (e.g., BM regeneration after consolidation treatment). The date of recurrence is defined as the date of the first bone marrow test after CR/CRi consistent with disease recurrence; (ii) recurrence or development of cytologically proven extramedullary disease.

A patient was considered as primary treatment-refractory when failure to achieve a CR or CRi. Patients who required a second cycle of induction therapy to achieve CR/CRi were not considered to have refractory disease.

Main variables included sex, age, and clinical variables such as: clinically significant concomitant diseases (history of neoplasia or exposure to prior chemotherapy/radiotherapy), main baseline characteristics of patients at the time of AML diagnosis (laboratory data, FAB subtype, cytogenetic [15] and molecular characteristics [16]) when available; prior treatments received and responses obtained; number of previous lines for AML before starting venetoclax treatment; start and end dates of venetoclax treatment, administered dose and whether the combination drug was either HMA or LDAC. When available, the next-generation sequencing (NGS) profile was included.

The primary end-point was CR/CRi, according to Cheson et al. response standardization criteria [17], and OS in all reported patients. Response to venetoclax combination treatment was evaluated under the 2017 ELN response standards [16].

To assess safety, the incidence of grade III/IV adverse events were registered when documented in the patient’s medical records, discontinuation of venetoclax due to severe toxicity, as well as days and number of hospitalizations were also accounted for.

### 2.5. Statistical Analysis

We used Microsoft Excel for registering clinical data. Results were presented as percentages for categorical variables and as medians (and range) for continuous variables. We used the IBM SPSS Statistics package, version 19.0 (SPSS, Chicago, IL, USA), to assess differences between groups. The chi-squared test or the Fisher exact test were used for the comparison of categorical variables. The overall survival was calculated using the Kaplan–Meier method, defining the time-to-event as the time from initiation of venetoclax to the last follow-up or death from any cause. The log-rank test was used to compare survival curves. All the variables that showed statistical significance in the univariate analyses were included in a multivariate analysis using the Cox proportional hazard method. All *p* values reported were 2-sided, and statistical significance was set at *p* < 0.05.

## 3. Results

### 3.1. Patient’s and Disease Characteristics

A total of 51 patients were identified, 33 men and 18 women, with a median age of 68 years (ranging from 25 to 82 years old) and an ECOG ≥ 2 at the beginning of the venetoclax treatment in 52% of the cases. Table 1 shows the main baseline characteristics of the study population at diagnosis and prior to venetoclax treatment. Coagulation, liver enzymes and renal function patients’ characteristics can be found in Appendix A. Two-thirds of the patients had AML with myelodysplasia-related changes (AML-MRC). A total of 22 (52.4%) had an adverse cytogenetic profile according to Medical Research Council [15], and 31 patients were high risk according to European LeukemiaNet 2017 criteria [16].

Nine patients (18%) received a previous HSCT in the first line, and three patients (6%) received an allo-SCT in subsequent lines prior to venetoclax therapy. A total of 61% of patients received ≥2 previous lines for AML (range, 1–4), and 26 patients (51%) had received HMAs prior to venetoclax. A total of 22 patients were primary refractory to first-line therapy, but 30 were refractory to any line prior to venetoclax (58.8%), including 17 patients who failed to achieve a CR/CRi with HMAs (33.3%).

The median time from AML diagnosis to the first dose of venetoclax was 10 months (range, 1–65). A total of 35 patients (68.6%) received intensive chemotherapy at the first line (mostly 3 + 7), followed by HMAs (23.5%) and 4 patients received an LDAC-based regimen (usually LDAC and oral fludarabine with G-CSF, FLUGA regimen [18]). Previous treatment patterns for different venetoclax combinations are shown in Table 1. First-line therapies differed between patients salvaged with venetoclax and LDAC (33.3% received prior intensive chemotherapy and 50% HMAs) compared with venetoclax and HMAs (73.3%. received prior intensive chemotherapy and 20% HMAs). Overall, 25 patients (51%) achieved CR/CRi after the first-line treatment, 16% had a PR, and 33% primarily failed to respond.

All patients salvaged with venetoclax plus LDAC had previously received at least one line containing HMAs, compared with 60% of patients with venetoclax + decitabine and 33.3% of patients with venetoclax + azacitidine.

#### Venetoclax Treatment

With a median follow-up of 167 days (range, 21–311) from the first dose of venetoclax, the median venetoclax treatment duration was 49 days, with 10 out 51 (19%) patients still on venetoclax at the last follow-up.

Venetoclax was administered with AZA in 59% of patients, with decitabine (DEC) in 29%, and with LDAC in 12%. Patients received a median of 2 cycles (range, 0–8). One patient died before completing the first venetoclax cycle, 21 patients (41%) received 1 cycle, 13 received a second cycle (25.5%) and 16 patients received ≥ 3 cycles (31.5%).

The dose ranged from 70 to 800 mg. A total of 29 out of 46 patients with available data received concomitant medication with CYP3A4 inhibitors: ciprofloxacin (*n* = 5), clotrimazol (*n* = 2), fluconazole (*n* = 6), posaconazole (*n* = 17), voriconazole (*n* = 5), and isavuconazole (*n* = 1), and 22/29 patients received reduced doses of venetoclax due to the concomitant treatment.

### 3.2. Response Rates

The overall CR rate was 10.4% (*n* = 5), CRi 2% (*n* = 1), PR 10.4% (*n* = 5), 46% resistance (*n* = 22), 15 patients died before response assessment (31%), and response was not available in 3 alive patients.

The CR/CRi and ORR (CR/CRi+PR) rate were 17.9% (*n* = 5/28) and 32.1% (*n* = 9/28) in patients who received venetoclax plus AZA, 6.7% (*n* = 1/15) and 13.3% (*n* = 2/15) in those who received decitabine and 0/5 in patients who received LDAC plus venetoclax.

The only variables associated with higher CR/CRi were NPM1 mutated, 3/6 (50%) vs. 3/33 (9.1%) compared with NPM1 negative, *p* = 0.036, and CEBPA single or double mutated, 2/3 (66.7%) vs. 1/20 (5%) compared with CEBPA negative, *p* = 0.034. CEBPA mutated was also associated with higher ORR (3/3) than CEBPA wild type (*n* = 6/33, 18.2%), *p* = 0.002. (Appendix A).

### 3.3. Subsequent Salvage Therapy

A total of 12 patients received a subsequent salvage therapy after venetoclax, 10 out of 22 resistance patients and 2/5 patients who achieved PR. Most of these patients received intensive therapy (10/12), and 5 of them received an allogeneic-SCT. Responses after subsequent therapies were assessed in 11 patients: 4 achieved CR, 5 were resistant, and 1 patient died.

### 3.4. Overall Survival

With a median follow-up time of 166 days (range, 21–311), 66.7% of the patients died. The estimated median OS from venetoclax initiation was 104 days (95% CI: 56–151) (Figure 1). The OS was 120 days (95% CI: 77–163) when venetoclax was administered in combination with azacitidine; 104 days (95% CI: 0–244) when administered with decitabine and 69 days (95% CI: 56–81) when administered with LDAC; *p* = 0.875.

Those responder patients had higher OS, 215 days for CR/CRi vs. 144 days (95% CI: 120–167) for PR vs. 69 days (95% CI: 54–83) for non-responder patients (*p* = 0.008) (Figure 2). Baseline characteristic influencing OS was ECOG 0 when starting venetoclax, with median OS not reached vs. 75 days (95% CI: 35–114) in those with ECOG ≥1, *p* = 0.001 (Appendix A). In a multivariate model including sex and age (Table 2), ORR (CR/CRi+PR), ECOG, and sex retained its statistical significance with OS.

Finally, in those patients who were resistant to venetoclax (*n* = 22), the median OS after cessation of venetoclax therapy was 42 days (CI 95%: 0–86). Those resistant patients who could receive a subsequent salvage therapy had superior OS (98 vs. 5 days, *p* = 0.004).

### 3.5. Toxicity

A total of 28% of the patients required discontinuation of treatment due to toxicity. A total of 61% of the patients were admitted at some time during treatment with venetoclax. Infections were the most frequent reason for admission. Up to 27 patients developed an infectious episode (mainly febrile neutropenia), and 9 patients died. Gram-negative bacteria, such as *Escherichia coli* and *Klebsiella pneumonia*, were the most frequent microbiological documentation when available. Other reasons for admission were bleeding (10%) and other causes (12%). Only one case of tumor lysis syndrome was described.

## 4. Discussion

This study describes the outcome of a series of R/R-AML patients treated with venetoclax in combination with other drugs. Patients in this cohort, belonging to the PETHEMA registry, showed decreased responses rates and lower overall survival compared to most of retrospective case series of R/R-AML patients salvaged treated with venetoclax-based regimens [10,12,13,19,20,21]; yet they were similar to those from an R/R-AML phase 1 trial with comparable baseline characteristics [11].

Our real-life series depicts the marginal probability of CR/CRi (12%) and an OS (median, 104 days) after venetoclax-based salvage. Early monotherapy data from a phase II trial by Konopleva et al. [22] showed a CR/CRi rate of 19% in R/R-AML patients. Several studies reported results using venetoclax-based combinations in patients with R/R-AML [10,11,12,13,23], with a range of CR/CRi rates between 10% and 50% and median OS between 3 and 6.6 months. In our study, even poorer outcomes were obtained. Possible explanations for differences in results with published literature could be: (1) Our series included heavily pretreated patients (i.e., roughly half of them had been previously treated with HMAs, one-third were considered primary refractory to HMAs, and 60% were refractory to any prior line); (2) there might be a potential selection bias in our registry toward bad outcomes patients reported; or conversely, a selection bias in other registries toward suitable outcomes patients reported.

A recent single-institution study, which included 14 R/R-AML patients with a median age of 58 years, showed a CRi rate of 21.4% (no CR were observed) and a median OS of 4.7 months [13]. Aldoss et al. retrospectively enrolled 33 patients with R/R-AML treated with venetoclax and HMAs combination, showing a CR/CRi rate of 46%, but in a lower risk cohort [20]. In our series, 86% had high risk according to the ELN2017. In line with our study, DiNardo et al. found a CR/CRi rate of 12%, and median overall survival was 3.0 months [11]. They included an older (median, 68 years) and poor prognosis population of 43 patients, among which all but 2 had adverse or intermediate cytogenetics [11].

Previous retrospective studies in R/R-AML patients treated with venetoclax combination therapy showed that the best response was achieved after a median of 2 cycles (range 1–3) [11,24]. However, most of our patients (70%) received only 1 or 2 venetoclax cycles, probably contributing to the low CR/CRi rate herein reported. In this context, it is still necessary to identify those patients at higher risk of early death, and therefore, that will not benefit from this type of treatment. For these patients, other treatment strategies could be recommended.

Data on predictors of outcomes of venetoclax therapy in R/R-AML are very limited. Aldoss et al. described a tendency toward a higher response rate in de novo and therapy-related AML compared to secondary AML in R/R-AML [24]. In our study, 13 patients had a secondary AML, and none of them achieved CR/CRi. The only variables associated with higher CR/CRi were NPM1 mutated (50%), and CEBPA mutated (66.7%). Similarly, in a recent retrospective study in R/R-AML patients treated with venetoclax, NPM1 mutations was a significant predictor of response (with CR/CRi of 46%), and CEBPA mutations were also associated with higher CR/CRi [21]. In our study, other molecular mutations such as IDH1/2, P53, or FLT3-ITD mutations did not influence response. We also show that in the R/R-AML, the ECOG 0 and response to therapy were associated with significantly superior survival (median not reached, and 7 months, respectively). ECOG performance status has been described as one of the main predictors for early death in unfit AML patients who were treated in front-line with HMAs [25], but it was not evaluated in this context. Finally, we also observed a longer overall survival among women treated with venetoclax. This is not a surprising finding as several reports have shown worse outcomes in male AML patients [26,27]. Although the mechanism explaining this difference is unclear, it could hint at the possible role of hormonal variations in the biology of the disease and possible social influences [26].

We could not demonstrate statistically significant differences in terms of response or survival among patients treated with venetoclax + AZA or DEC or LDAC, but patients treated with LDAC and DEC had lower median OS in line with prior observations [21]. As in the series by Stahl et al., in our study, prior treatment with hypomethylating agents did not influence response or survival in R/R-AML treated with venetoclax combinations, even in the group of AZA + venetoclax [21].

Further perspectives include increasing our sample size so that more solid conclusions can be drawn. Therefore, further research is warranted.

## 5. Conclusions

This real-life series depicts a marginal rate of CR/CRi and poor OS following venetoclax-based salvage therapy. Patients in this cohort exhibited high-risk features, which could partly explain the observed poor outcomes. Acknowledging the limited follow-up, the rather small proportion of responders did not reach the median OS. Further studies will help to identify what subset of R/R-AML patients will potentially benefit from venetoclax-based salvage schemes. At present, enrollment in clinical trials remains the preferred key option for the management of these patients.

## Figures and Tables

**Figure 1 cancers-14-01734-f001:**
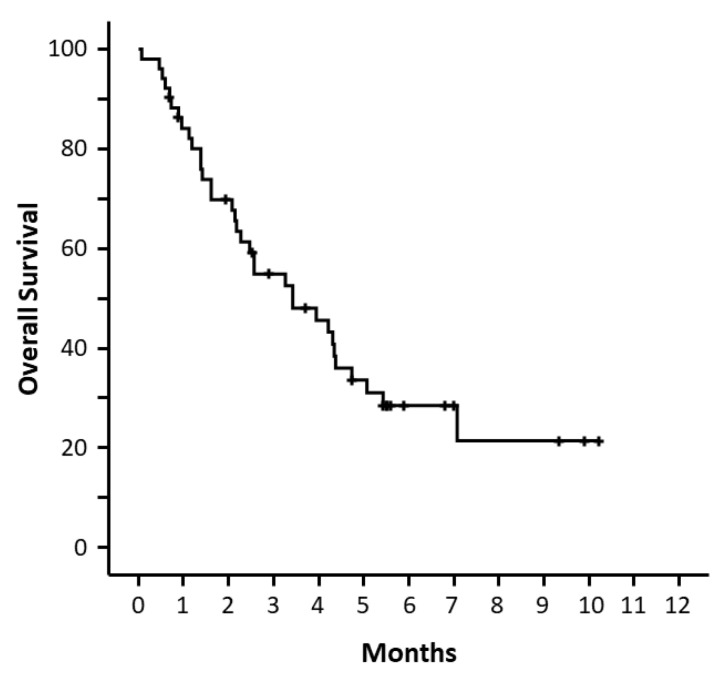
Overall survival from the start of venetoclax.

**Figure 2 cancers-14-01734-f002:**
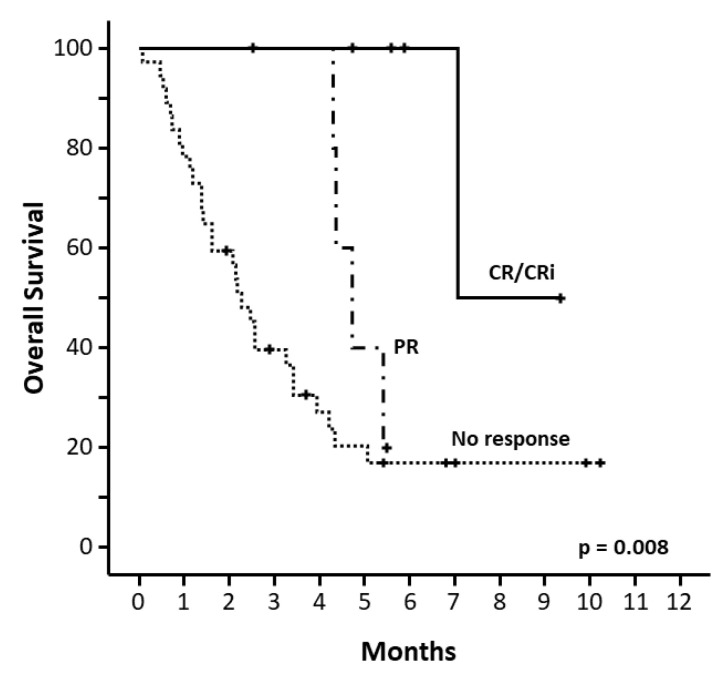
Overall survival from the start of venetoclax according to response.

**Table 1 cancers-14-01734-t001:** Main patients’ characteristics prior to venetoclax treatment.

Variable	All Patients		Azacitidine + Venetoclax		Decitabine + Venetoclax		Low-Dose Cytarabine + Venetoclax	
	*n* (%)	Median (Range)	*n* (%)	Median (Range)	*n* (%)	Median (Range)	*n* (%)	Median (Range)
Total	51 (100)		30 (58.8)		15 (29.4)		6 (11.8)	
Sex (male), *n* (%)	33 (64.7)		19 (63.3)		10 (66.7)		4 (66.7)	
Age, median (range)	51	68 (25–82)	30	67 (41–76)	15	64 (25–82)	6	74 (71–77)
≥65 y, *n* (%)	34 (66.6)		21 (70)		7 (46.7)		6 (100)	
Secondary AML	20/50 (40)		11/30 (36.7)		7/15 (46.7)		2/5 (40)	
MDS	13/20 (65)		6/		5		2	
MPNs	2/20 (10)		1		1		0	
t-AML	6/20 (30)		5		1		0	
ECOG performance status	50		30		9		5	
0	10 (20)		7 (23.3)		5 (55.6)		3 (60)	
1	14 (28)		18 (60)		3 (33.3)		2 (40)	
≥2	26 (52)		5 (16.7)		1 (11.1)		0	
WBC count (×10^9^/L)	48	2 (0–640)	29	3.0 (0–110)	13	1.0 (0–640)	6	3.5 (1.0–20)
Hemoglobin (g/dL)	48	8.0 (6–13)	29	9.0 (7.0–13)	13	8.0 (6.0–10.0)	6	8.5 (7.0–10.0)
Bone marrow blast count, %	44	36.5 (0–95)	28	29 (2–92)	10	45 (0–95)	6	39 (24–86)
≥50%	16 (36.4)		8 (28.6)		5 (50)		3 (50)	
FAB subtype	43		30		15		6	
M0/M6/M7	9 (21)		5 (16.7)		4 (26.7)		0	
M1/M2	10 (23)		5 (16.7)		3 (20)		2 (33.3)	
M4/M5	11 (25.5)		6 (20)		3 (20)		2 (33.3)	
Other	13 (30)		14 (19.8)		5 (33.3)		2 (33.3)	
Myelodysplasia-related changes AML	34 (66.7)		19 (63.3)		12 (80)		3 (50)	
Cytogenetics	44		26		13		5	
Favorable/Intermediate	24 (54.5)		14 (53.8)		6 (46.2)		4 (80)	
Adverse	20 (45.5)		12 (36.2)		7 (53.8)		1 (20)	
MRC risk stratification	42		26		12		4	
Favorable/intermediate	20(47.6)		12 (46.2)		5 (41.7)		3 (75)	
Adverse	22 (52.4)		14 (53.8)		7 (58.3)		1 (25)	
ELN 2017 risk stratification	36		23		10		3	
Favorable/intermediate	5 (14)		3 (139)		1 (10)		1 (33.3)	
Adverse	31 (86)		20 (87)		9 (90)		2 (66.7)	
Somatic mutations								
NPM1	6/41 (15)		3/27 (11.1)		2/11 (18.2)		1/3 (33.3)	
FLT3-ITD	5/41 (12)		4/27 (14.8)		1/11 (9.1)		0/3 (0)	
P53	8/29 (27)		6/20 (30)		2/7 (28.6)		0/2 (0)	
IDH1/2	9/27 (29)		6/18 (33.3)		1/7 (14.3)		1/2 (50)	
First-line treatment								
Intensive chemotherapy, *n* = 35	35 (69)		22 (73.3)		11 (73.3)		2 (33.3)	
LDAC-based regimen, *n* = 4	4 (8)		2 (6.7)		1 (6.7)		1 (7.8)	
HMAs, *n* = 12	12 (23)		6 (20.0)		3 (20)		12 (23.5)	
HMAs at any line prior to venetoclax, *n* = 26	26 (51)		10 (33.3)		10 (66.7)		6 (100)	
Previous stem cell transplant, *n* = 12	12 (23)		8 (26.7)		4 (28.6)		0 (0)	
Median number of previous lines (range)				1 (1–4)		2 (1–4)		2 (1–3)
AML status								
Refractory	22 (43.1)		16 (53.3)		5 (33.3)		1 (16.7)	
Relapse 1	23 (45.1)		12 (40.0)		8 (53.3)		3 (50.0)	
Relapse ≥ 2	6 (11.8)		2 (6.7)		2 (13.3)		2 (33.3)	
Refractory to any line prior VEN	30 (58.8)		18 (60.0)		10 (66.7)		2 (33.3)	
Refractory to prior HMAs	17 (33.3)		7 (23.3)		8 (53.3)		2 (33.3)	

Abbreviations: AML, acute myeloid leukemia; ELN, European LeukemiaNet; ECOG, Eastern Cooperative Oncology Group scale; FAB, French-American-British; FLT3-ITD, fms-related receptor tyrosine kinase 3 internal tandem duplications; HMAs, hypomethilating agents; LDAC, low-dose cytarabine; MDS, myelodysplastic syndrome; MPNs, myeloproliferative neoplasms; MRC, Medical Research Council; NMP1, nucleophosmin 1; t-AML, therapy-related acute myeloid leukemia; VEN, venetoclax; WBC, white blood cell.

**Table 2 cancers-14-01734-t002:** Multivariate analysis of factors influencing survival in R/R-AML patients treated with venetoclax.

Variable	Median (Days)	P-Univariate	P-Multivariate	HR (95% CI)
Total, *n* = 47	78			
Age				
<65 y, *n* = 17	34	0.919	0.326	1.48 (0.68–3.22)
≥65 y, *n* = 34	34			
Sex				
Female, *n* = 18	131	0.217	0.010	3.37 (1.45–7.82)
Male, *n* = 33	78			
ECOG performance status				
0, *n* = 10	NR	0.001	0.005	14.96 (1.91–117.21)
≥1, *n* = 40	75			
ORR (CR + CRi + PR)				
Yes,11	215	0.004	0.002	6.13 (0.68–3.22)
No, 37	69			

Abbreviations: CR, complete remission; CRi, CR with incomplete blood count recovery; ECOG, Eastern Cooperative Oncology Group scale; PR, partial response; R/R-AML, relapsed/refractory acute myeloid leukemia.

## Data Availability

The data presented in this study are available on request from the corresponding author. The data are not publicly available due to privacy concerns.

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
