# Peer review of "Use of Venetoclax in Patients with Relapsed or Refractory Acute Myeloid Leukemia: The PETHEMA Registry Experience"

_cancers, 2022, doi:10.3390/cancers14071734_

Round 1

Reviewer 1 Report

In the manuscript with the title „Use of Venetoclax (VEN) in Patients with relapsed or Refractory Acute Myeloid leukemia: the Pethema registry Experience” Jorge Labrador et al. report the effectiveness of VEN in a cohort of 51 RR-AML patients using information from a multicenter observational database. In addition, predictors of response and survival were evaluated. VEN was used in 30 patients in combination with Azacitidine (AZA), in 15 patients in combination with Decitabine (DEC) and in 6 patients with low-dose araC (LDAC). The patients received intensive chemotherapy (n=35), HMAs at any line (25%), HMAs (n=12), previous HSCT (n=12) and LDAC-based regimens (n=4). Patients treated with VEN+AZA had CR+CRi in 17.9%, with VEN+DEC in 6.7% and with VEN+LDAC in 0%. Despite having received a median of 2 VEN cycles, 41% received only one cycle, 25.5 two cycles and 31.5% ≥3 cycles. Median OS for all patients amounted to 104 days, 120 days in combination with AZA, 104 days in combination with DEC and 69 days with LDAC. Treatment response and ECOG 0 (and gender) influenced OS in a multivariate analysis. A considerable amount of patients (28%) required discontinuation of VEN due to toxicity. The authors conclude that after VEN based salvage there is a marginal probability of CR/CRi and poor OS.

This is an interesting real-life retrospective study with responses inferior to most of the retrospective study, but similar to the phase I study in R/R

Comments

The patients treated with VEN are characterized according to extensive parameters, but not regarding disease status at VEN start. How many of these patients were refractory and how many were relapsed? This information is not given in Table 1 and might be important for understanding differences to published studies.

Table 1 reports many characteristics before VEN treatment including liver enzymes, renal function and molecular markers. The information here should restricted to the important variables and AML characteristics should refer to the time points at diagnosis and at VEN start. Molecular data were determined probably at diagnosis? The major point relates to the treatment before VEN treatment. It may make a difference if the patient were refractory to AZA before starting VEN+AZA. This information should be given in Table 1 in more detail. I suggest to join Table 1+2 and remove not important information on Table 1.

Definition of relapse is given as increase in blasts after CR, but the term refractory needs further explanation. How many induction cycles were given to refractory patients or are the refractory patients only those not responding to chemotherapy (primary refractory)? The authors may want to split patients in Table 1 in refractory and relapsed.

In the discussion, I miss possible explanations for differences in results with published literature. Age is mentioned, but more high-risk AML might prevail (how many were resistant to AZA and received VEN+AZA). “Heavily pretreated” is not enough. I would also be more cautious in interpreting results on 6 patients! Furthermore less treatment cycles may cause the inferior results. These are important information for the reader.

What kind of infections were observed under VEN treatment? 45% of the patients stopped VEN treatment because of (life-threatening?) infections.  

Gender seems also to be a significant variable in the multivariate analysis. Should be mentioned in the manuscript and in the discussion section.

Line 295 “) and. CEPBA”?

Reference 16 and 18 is redundant

Author Response

Reviewer #1:

In the manuscript with the title „Use of Venetoclax (VEN) in Patients with relapsed or Refractory Acute Myeloid leukemia: the Pethema registry Experience” Jorge Labrador et al. report the effectiveness of VEN in a cohort of 51 RR-AML patients using information from a multicenter observational database. In addition, predictors of response and survival were evaluated. VEN was used in 30 patients in combination with Azacitidine (AZA), in 15 patients in combination with Decitabine (DEC) and in 6 patients with low-dose araC (LDAC). The patients received intensive chemotherapy (n=35), HMAs at any line (25%), HMAs (n=12), previous HSCT (n=12) and LDAC-based regimens (n=4). Patients treated with VEN+AZA had CR+CRi in 17.9%, with VEN+DEC in 6.7% and with VEN+LDAC in 0%. Despite having received a median of 2 VEN cycles, 41% received only one cycle, 25.5 two cycles and 31.5% ≥3 cycles. Median OS for all patients amounted to 104 days, 120 days in combination with AZA, 104 days in combination with DEC and 69 days with LDAC. Treatment response and ECOG 0 (and gender) influenced OS in a multivariate analysis. A considerable amount of patients (28%) required discontinuation of VEN due to toxicity. The authors conclude that after VEN based salvage there is a marginal probability of CR/CRi and poor OS.

This is an interesting real-life retrospective study with responses inferior to most of the retrospective study, but similar to the phase I study in R/R

 Comments

The patients treated with VEN are characterized according to extensive parameters, but not regarding disease status at VEN start. How many of these patients were refractory and how many were relapsed? This information is not given in Table 1 and might be important for understanding differences to published studies.

Response: Thank you for your comment. We agreed with the reviewer and added the information in table 1.

Table 1 reports many characteristics before VEN treatment including liver enzymes, renal function and molecular markers. The information here should restricted to the important variables and AML characteristics should refer to the time points at diagnosis and at VEN start. Molecular data were determined probably at diagnosis? The major point relates to the treatment before VEN treatment. It may make a difference if the patient were refractory to AZA before starting VEN+AZA. This information should be given in Table 1 in more detail. I suggest to join Table 1+2 and remove not important information on Table 1.

Response: Thank you for your comment. We agreed with the reviewer#1 and reviewer#2 and deleted the information regarding coagulation and liver and renal function from table 1 and placed it as supplementary table 1. Moreover, following the reviewer suggestion, table 2 has been merged with table 1. Furthermore, information regarding patients refractory to any prior line before VEN and refractory to HMA has been added.

Definition of relapse is given as increase in blasts after CR, but the term refractory needs further explanation. How many induction cycles were given to refractory patients or are the refractory patients only those not responding to chemotherapy (primary refractory)? The authors may want to split patients in Table 1 in refractory and relapsed.

Response: thank you for your suggestion. The definition of refractory has been added into methods section, as follows: “A patient was considered as primary treatment-refractory when failure to achieve a CR or CRi. Patients who required a second cycle of induction therapy to achieve CR/CRi were not considered to have refractory disease.”

In the discussion, I miss possible explanations for differences in results with published literature. Age is mentioned, but more high-risk AML might prevail (how many were resistant to AZA and received VEN+AZA). “Heavily pretreated” is not enough. I would also be more cautious in interpreting results on 6 patients! Furthermore less treatment cycles may cause the inferior results. These are important information for the reader.

Response: thank you for your suggestion. We have added the following into discussion: “Possible explanations for differences in results with published literature could be: 1) Our series included heavily pretreated patients (i.e, roughly half of them had been previously treated with HMAs, one third were considered primary refractory to HMAs, and 60% were refractory to any prior line; 2) there might be a potential selection bias in our registry towards bad outcomes patients reported; or conversely, a selection bias in other registries towards good outcomes patients reported.”

What kind of infections were observed under VEN treatment? 45% of the patients stopped VEN treatment because of (life-threatening?) infections.  

Response: Thank you for your question. We have clarified in the results section “Twenty-eight percent of the patients required discontinuation of treatment due to toxicity. Sixty one percent of the patients were admitted at some time during treatment with venetoclax. Infections were the most frequent reason for admission. Up to 27 patients developed an infectious episode (mainly febrile neutropenia) and nine patients died. Gram-negative bacteria, such as Escherichia coli and Klebsiella pneumonia, were the most frequent microbiological documentation, when available. Other reasons for admission were bleeding (10%) and other causes (12%). Only one case of tumor lysis syndrome was described.”

Gender seems also to be a significant variable in the multivariate analysis. Should be mentioned in the manuscript and in the discussion section.

Response: thank you for your suggestion. We have added the following into discussion “Finally, we also observed a longer overall survival among women treated with venetoclax. This is not a surprising finding as several reports have showed worse outcomes in male gender AML. Although the mechanism explaining this difference is unclear, it could hint at the possible role of hormonal variations in the biology of the disease and possible social influences.”

Line 295 “) and. CEPBA”?

Response: thank you for noticing and forgive the mistake, which has been corrected.

Reference 16 and 18 is redundant

Response: thank you for noticing and forgive the mistake, which has been corrected.

Reviewer 2 Report

Labrador et al described the result of retrospective survey administrated VEN combined with AZA/DEC or LDAC for refractory or relapsed AML.

Currently, no results are reporting prospective comparisons between VEN combined with AZA/DEC or LDAC. The results of this retrospective study will give a novel insight for further development of randomized study using VEN combined with AZA/DEC or LDAC in the future.

Major concerns

Although the result of multivariate analysis is described in Table 3, dosed this analysis include all variates described in Table 1? The reviewers feel that some of the parameters listed in Table 1 are biased.

Minor concerns

Table 1

The information, Platelet < 20 x109 /L, PT or APTT, Creatinine, Uric acid, Bilirubin, AST, ALT, ALP, LDH, Albumin should be provided as supplemental data.

Secondary AML

In all patients, there is 4 lines, however, in Azacitidine + venetoclax, Decitabine + venetoclax, there are 5 lines. Low-dose cytarabine + venetoclax, there are 3 lines.

ECOG PS

In all patients, there is 6 lines, however, in Azacitidine + venetoclax, Decitabine + venetoclax, Low-dose cytarabine + venetoclax, there are 5 lines.

FAB subtype

In all patients, there is 5 lines, however, in Azacitidine + venetoclax, Decitabine + venetoclax, Low-dose cytarabine + venetoclax, there are 4 lines.

Please carefully double-check the data in Table 1.

Author Response

Reviewer #2:

Labrador et al described the result of retrospective survey administrated VEN combined with AZA/DEC or LDAC for refractory or relapsed AML.

Currently, no results are reporting prospective comparisons between VEN combined with AZA/DEC or LDAC. The results of this retrospective study will give a novel insight for further development of randomized study using VEN combined with AZA/DEC or LDAC in the future.

Major concerns

Although the result of multivariate analysis is described in Table 3, dosed this analysis include all variates described in Table 1? The reviewers feel that some of the parameters listed in Table 1 are biased.

Response: Thank you for your comment. Multivariate analysis includes all variables described in table 1 in forwarded method that only include those that are associated with the outcome.

Minor concerns

Table 1

The information, Platelet < 20 x109 /L, PT or APTT, Creatinine, Uric acid, Bilirubin, AST, ALT, ALP, LDH, Albumin should be provided as supplemental data.

Response: thank you for your suggestion. We have placed that information as supplementary table 1.

Secondary AML

In all patients, there is 4 lines, however, in Azacitidine + venetoclax, Decitabine + venetoclax, there are 5 lines. Low-dose cytarabine + venetoclax, there are 3 lines.

 ECOG PS

In all patients, there is 6 lines, however, in Azacitidine + venetoclax, Decitabine + venetoclax, Low-dose cytarabine + venetoclax, there are 5 lines.

 FAB subtype

In all patients, there is 5 lines, however, in Azacitidine + venetoclax, Decitabine + venetoclax, Low-dose cytarabine + venetoclax, there are 4 lines.

 Please carefully double-check the data in Table 1.

Response: thank you for your suggestion. We have revised table 1 and corrected mistakes when noticed.